# The Dynamics of Nerve Degeneration and Regeneration in a Healthy Milieu and in Diabetes

**DOI:** 10.3390/ijms242015241

**Published:** 2023-10-16

**Authors:** Lars B. Dahlin

**Affiliations:** 1Department of Translational Medicine—Hand Surgery, Lund University, SE-205 02 Malmö, Sweden; lars.dahlin@med.lu.se; Tel.: +46-40-33-17-24; 2Department of Hand Surgery, Skåne University Hospital, SE-205 02 Malmö, Sweden; 3Department of Biomedical and Clinical Sciences, Linköping University, SE-581 83 Linköping, Sweden

**Keywords:** nerve degeneration, nerve regeneration, nerve injury, diabetes, neuropathy, axonal outgrowth, neuroprotection, apoptosis, nerve repair, nerve reconstruction, proteomics, synchrotron, nanotomography

## Abstract

Appropriate animal models, mimicking conditions of both health and disease, are needed to understand not only the biology and the physiology of neurons and other cells under normal conditions but also under stress conditions, like nerve injuries and neuropathy. In such conditions, understanding how genes and different factors are activated through the well-orchestrated programs in neurons and other related cells is crucial. Knowledge about key players associated with nerve regeneration intended for axonal outgrowth, migration of Schwann cells with respect to suitable substrates, invasion of macrophages, appropriate conditioning of extracellular matrix, activation of fibroblasts, formation of endothelial cells and blood vessels, and activation of other players in healthy and diabetic conditions is relevant. Appropriate physical and chemical attractions and repulsions are needed for an optimal and directed regeneration and are investigated in various nerve injury and repair/reconstruction models using healthy and diabetic rat models with relevant blood glucose levels. Understanding dynamic processes constantly occurring in neuropathies, like diabetic neuropathy, with concomitant degeneration and regeneration, requires advanced technology and bioinformatics for an integrated view of the behavior of different cell types based on genomics, transcriptomics, proteomics, and imaging at different visualization levels. Single-cell-transcriptional profile analysis of different cells may reveal any heterogeneity among key players in peripheral nerves in health and disease.

## 1. Introduction

A peripheral nerve trunk may be affected by various injuries, such as nerve compression, nerve crush, nerve transection, or nerve laceration, where the nerve fibers distal to the injury undergo Wallerian degeneration, i.e., the active process, involving a number of different cells and molecular mechanisms, that causes “anterograde” degeneration of the nerve fibers, with modifications in the local environment, after a nerve injury or in a neuropathy [1,2]. Similar Wallerian degeneration occurs in neuropathies, such as diabetic neuropathy, where axons and Schwann cells are affected by the different mechanisms of the underlying disease [1]. The biological phenomenon of axonal degeneration and regeneration in a peripheral nerve, which we normally relate to trauma of varying degrees, is fascinating and involves a variety of delicate and well-orchestrated mechanisms in the peripheral nerve [3,4] and in the central nervous system [5]. Axons attempt to regrow after an injury to restore functionality, and the efficiency, the direction, and their relation to other nearby located nerve fibers during the axonal regrowth may be similar irrespective of the why the axons are injured or affected by a disease. The outgrowth is dependent on the microenvironment (tissue niche) as well as on the different interacting cells, particularly the Schwann cells that are present in the local milieu where the axons grow [6,7,8,9,10,11,12,13]. The cleaning up of myelin debris, the modification of the microenvironment, and the interaction between the Schwann cells are particularly crucial for the nerve regeneration process [9,10,12,13]. These processes are also affected by aging [7] and the location of the injury in relation to the dorsal root ganglion (i.e., pre- or postganglionic injury [6]). When surgeons plan their strategies for nerve repair or nerve reconstruction of nerve injuries [11], they must have the detailed nerve regeneration mechanisms and potential applicable methods to stimulate nerve regeneration in mind [8,14,15,16,17,18,19]. Axons require a suitable and optimal environment to regenerate, irrespective of the occurrence of regeneration after an injury in a healthy subject or in a subject with diabetes, i.e., in a diabetic environment in an individual affected by a neuropathy. Importantly, axons do not regenerate in the milieu of the central nervous system despite their internal capacity, while the regeneration of axons in the peripheral nerve system is excellent under optimal conditions [20]. After the occurrence of cellular injury in a peripheral nerve, the denervated Schwann cells proliferate, elongate, and in some circumstances migrate as well as line up in the endoneurial tubes to support the outgrowing axons [21]. During this process, the Schwann cells up- and down-regulate different genes to adjust to their new situation where growth is prioritized compared to initial remyelination [20,21]. After the regenerating axons have reached their target(s), the Schwann cells down-regulate their grow-associated genes, and the axons relinquish their growth machinery returning the system to the “maintenance” condition [22].

The dynamic events of degeneration and regeneration in different neuropathies, where the axonal cytoskeleton is affected, e.g., in diabetic neuropathy [20], may be somewhat different in terms of the events after a trauma. The continuity of the nerve trunk is preserved in neuropathies despite the occurrence of axonal degeneration in the nerves affected by the underlying etiology. There may be an ongoing regenerative response associated with the formation of regenerative clusters of smaller myelinated nerve fibers and can be viewed as a morphometric shift towards smaller myelinated nerve fibers as observed in sural and posterior interosseous nerve biopsies from subjects with diabetes [23,24]. Such regenerative clusters are well-known entities, where axons can regenerate and originate from a node of Ranvier [25]. The observed degenerative phenomenon is often related and defined as axonal loss, but the segmental demyelination occurring in diabetic neuropathy should also be considered since the biological processes attempt to remyelinate axons after demyelination. This remyelination results in shorter internodes according to when nerves regenerate after a traumatic injury [9,26,27]. 

In the present review, degeneration and regeneration processes are focused on events occurring in healthy as well as in “diseased” microenvironments, using diabetes and an underlying diabetic neuropathy as experimental models [28]. In the presence of hyperglycemia, such as in type 1 diabetes in particular, the level of blood glucose may be important for the efficiency of both the degeneration and regeneration processes [29,30], but the lipid pattern (dyslipidemia) and obesity are also relevant mechanisms for the development of neuropathy and other nerve disorders in type 2 diabetes [31,32]. A moderately increased blood glucose level may have an impact on nerve degeneration and regeneration after surgical procedures of nerve injuries in some models of diabetes or diabetic neuropathies, and may have less impact on these processes in other models [30]. 

## 2. Axonal Outgrowth after a Traumatic Injury

The efficiency of axonal regeneration depends on a variety of factors, including the intrinsic capacity of neurons to regrow their axons after nerve injury, the rate of the neuronal synthesis of proteins and cytoskeletal organization, and the conditions of the environment in the denervated distal nerve stumps in which the axons intend to grow [20,33,34,35,36,37]. Contact attraction and contact repulsion and chemoattraction and chemorepulsion mechanisms direct the growth of the axonal growth cones in the extracellular matrix (ECM) at the site of injury and within the denervated nerve end [38]. The regulation of the ECM, and therefore the behavior of the denervated Schwann cells and the outgrowing axons, is extremely important as a mechanism in nerve regeneration [21,39,40]. An optimal supply of growth factors, after the clearance of the myelin debris in the denervated nerve end, is also important in the regeneration process, where augmentation of the growth hormone axis has been emphasized [41]. Many molecular pathways are involved in the neurons and in the Schwann cells for axonal outgrowth and regeneration [20,42]. 

After an injury, the distal part of the nerve fibers is separated from the rest of the neuron. These nerve fibers undergo Wallerian degeneration that involves the demyelination of the axons and the disintegration and fragmentation of the axons, when the myelin sheaths of the denervated Schwann cells are detached from the cells [1]. The clearance of the axonal and myelin debris is primarily a function of the macrophages that invade the nerve end, and the Schwann cells have been reported to mainly participate prior to their entry [43,44]. A cascade of events follows that promote axonal outgrowth with formation of growth cones that palpate the environment with their “fingers”, i.e., filopodia [45,46,47], attracted by factors as well as by the environment per se. In the peripheral nervous system, axonal outgrowth is usually excellent compared to the central nervous system, although the slow and sometimes misdirected axonal outgrowth depends on the type of injury. In the periphery, there are no barriers as in the central nervous system, where growth inhibiting molecules, such as NOGO-A, semaphorins, netrins, and ephrins, are present that inhibit axonal outgrowth [48]. However, the timing of any nerve repair or reconstruction is important so as to utilize the best “window of opportunity” [49,50,51] for the outgrowing axons with respect to the Schwann cell response and modification of the microenvironment [52], by fibronectin and different laminins [53,54].

After the nerve injury, different intrinsic mechanisms are stimulated in the neurons and in the Schwann cells, where the regeneration-associated transcription factors, such as c-Jun, activating transcription factor 3 (ATF3), cAMP response element-binding protein, signal transducer, activator of transcription-3, CCAAT/enhancer binding proteins β and δ, Oct-6, Sox11, p53, nuclear factor kappa-light-chain-enhancer of activated B cell, and ELK3, mediate the response [42]. Initially, there is an up-regulation of a variety of substances, including the expression of IL-6 (interleukin-6), leukemia inhibitory factor (LIF), and CNTF (ciliary neurotrophic factor) [55], in the neurons that activate other pathways. ATF3 is increased after the injury and is related to the increased expression of c-Jun-mediated regeneration-associated genes [8,56,57,58]. The expression of many other factors, including GAP-43, is increased as well [59]. The Ras/ERK pathway is one of the previously reported pathways since it involves nerve growth factor (NGF), which binds to the Trk receptor (tropomyosin receptor kinase) and activates Ras/ERK signaling. This pathway involves stimulation of Raf and MEK (MAP/ERK kinase) and finally extracellular signal-regulated kinase (ERK), which belongs to the MAPK (mitogen-associated protein kinase) family that is crucial for neurite outgrowth [60]. The activation of ERK in Schwann cells at the site of injury occurs very early after an injury [61,62] and is emphasized, among many other signal pathways, in different nerve injury/neuropathy models in which nerve regeneration appears [20]. Soluble neuregulin-1 has a pro-myelinating function in a dosage-dependent manner along with the involvement of the Ras/Raf/ERK pathway that drives the Schwann cell dedifferentiation [63,64,65]. The differences between peripheral and central nervous systems have been stressed [20]. Furthermore, the environment of a peripheral nerve is also efficient in promoting outgrowth of axons from the central nervous system, which has been described since long [66].

After a nerve injury, a fibrin matrix is rapidly formed between the two severed nerve ends. Schwann cells and axons, together with produced blood vessels, approach this matrix or a freeze-injured nerve environment [67,68,69]. The Schwann cells and axons grow in concert and interact closely [67,68,69], although the outgrowth is not perfectly directed and matched with their original endoneurial tubes. Formation of the matrix occurs if the distance between the nerve ends is not too long. The interaction-based growth of Schwann cells and axons occurs irrespective of the fibrin matrix, in an acellular nerve or in an acellular tendon autograft used for nerve regeneration [67,68,69]. Misdirected outgrowth of axons is also obvious in diabetic neuropathy, particularly in type 1 diabetes, which has been described in human nerve biopsies using nanotomography techniques (Figure 1 [25]).

Using this technique, one can track individual axons and their regrowth after degeneration and regeneration, visualizing that the axons are intertwingling between each other as “spaghetti”. Thus, such regeneration of two or several individual nerve fibers are clustered together [25], which can be seen as the well-known described regenerative clusters in 2D-light and electronmicroscopic images in nerve biopsies from humans with diabetes [24]. Details of “the birth of a new axon” from a node of Ranvier can also be visualized with nanotomography (Figure 2 [25]).

## 3. The Rapid Response in Nerves to Injury

After a nerve injury, there are rapid changes in the axons and in the Schwann cells with the upregulation of the transcription factors [70,71]. These changes convert the cells from a maintenance to a production machinery for regeneration, where the Schwann cells attract the axons to undergo regeneration (Figure 3). The wide range of changes that occur in the injured neurons have been described extensively in reviews [72,73]. The prompt response after the nerve injury includes calcium influx into the axoplasm, which is a fast and clear signal of a response to the injury, to elicit several events involving the cellular machinery to regrow axons. The intracellular calcium levels activate downstream effectors for the initiation of the regeneration process [72]. There are also alterations in anterograde and retrograde axonal transports along the microtubules that are crucial for the regenerative response, where the transport is dynein-dependent, i.e., transport that is affected in disease, such as diabetes [74].

One of the steps is also the effect of importin-*a* and importin-ß that constitute the nuclear import complex (NIC), which is crucial for eliciting the nuclear changes needed for executing the regenerative response. The detailed and likely complex responding mechanisms, including the signals elicited by the NIC, are probably disturbed in diabetes where nerve regeneration may be impaired. This impairment has been observed in different animal models [29,30,74]. The retrograde signaling also includes retrograde transport of ERK up to the nerve cell body, being differentially activated in diabetes [61], and the retrograde transport is importin-mediated [75]. One of the key differences between a central and a peripheral response to a nerve injury is the formation of growth cones by the peripheral axon in a suitable environment, compared to an inappropriate, i.e., negative, environment in the CNS shortly after the nerve injury. The assembly of an effective growth cone machinery is also dependent on the calcium influx. The basic cytoskeletal dynamics after nerve degeneration and regeneration has been investigated extensively in hippocampal neurons in culture, describing the formation of growth cones following axotomy [76,77,78], and is also applicable for peripheral axons [79]. However, the prerequisite of an optimal nerve regeneration is the creation of an appropriate environment in the peripheral nerve, allowing the axons with their sprouts and growth cones to advance in the correct direction towards their targets. The key players are the Schwann cells, the macrophages, fibroblasts, and the extracellular matrix with their secreted signals (i.e., secretomes [80,81]) that contain proteins [82], which are also key players in diabetes [10,83,84,85].

There are numerous regeneration-associated genes (RAGs) that are activated in peripheral nerves after injury as a response. The different transcriptional targets and specific functions of these factors have not been fully clarified [86]. Among the transcription factors that are associated with regeneration, Jun [87,88,89] as well as ERK [61] are rapidly up-regulated after a nerve injury. It must be stressed that there is a difference in the up-regulation of genes between sensory and motor neurons, for example, motor neurons lack Jun and thus an alternative SRF-dependent gene expression program is initiated [86]. These gene expressions are relevant as they include plasticity-associated transcription factors that may lead to an aberrant early increasing synapse density of motor neurons. Thus, it has been emphasized that Jun is relevant soon after nerve injury since it pushes the injured neurons away from plasticity response and towards a phenotype of regeneration [86].

Regeneration is initiated by the upregulation of around 254 essential genes in a variety of functional categories after an axotomy of facial motor neurons [86]. Furthermore, ATF3 is important for the activation of regenerative gene transcription in sensory neurons [90]. Many functional clusters of genes that are related to the promotion or the repression of axonal outgrowth, which also includes the guidance of axons, involving microtubule dynamics of relevance for growth cone formation and thereby axonal extension, have been defined [91,92]. Furthermore, the gene expression also includes genes relevant to the extracellular matrix components as well as a few genes that have an inhibitory function on axonal outgrowth [93]. Several genes may be required for axonal outgrowth after injury but may not be of relevance during development [92,94]. The events occurring with regard to gene expression in the neurons and other cells that are key players after nerve injury and repair or reconstruction are not fully clarified, despite reported pervasive changes in gene expression related to the mentioned growth cone formation and axonal guidance [92,95]. Future research will hopefully further explain the relevance of these events and their impact on treatment of nerve injuries, possibly as a part of “precision medicine”.

## 4. The Schwann Cell after Injury

Schwann cells have a unique capacity to support axonal outgrowth and guide the axons down appropriate pathways and targets at different levels along the injured nerve, such as at the site of injury, along the injured distal nerve end, and close to the neuromuscular junction, where the terminal Schwann cells are located [10,96] (Figure 4). The process requires efficient clearance of the myelin debris after the nerve injury. Inefficient clearance of such debris, due to the inhibition of macrophage invasion, can be an obstacle for axonal outgrowth [97]. The clearance has also been described as a reason for poor regeneration in aging animals together with the Schwann cell response [73,98,99].

Myelinating Schwann cells do not turn over in adult nerves, but dedifferentiate after nerve injury and convert to the proliferating progenitor-like cells that orchestrate the regenerative response [100] (Figure 4). The progenitor-like Schwann cells, or repair Schwann cells, have a unique phenotype although their functions have still not been revealed completely [101]. The analysis of the transcriptome, also investigated at the subcellular level (subcellular map) [42,102], shows the signature of messenger RNA (mRNA) molecules in the Schwann cells that are related to repair (“repair Schwann cells”), compared to those that are not related to repair (“non-repair Schwann cells”) and are clearly different. The genes in “repair Schwann cells” that are related to inflammation, repair, and regeneration are up-regulated [101]. In contrast, the “non-repair Schwann cells” seem to up-regulate genes associated with the maintenance of myelin [101]. After the dedifferentiation and during the regeneration process, the Schwann cells re-differentiate to new functional myelin-forming cells, including also becoming non-myelinating Schwann-cells [100].

The amazing plasticity of denervated Schwann cells to adapt to the new situation, irrespective of being myelinating and non-myelinating (Remak) cells, should be highlighted, particularly concerning the large and small diameter axons and their regenerative capacity. The Schwann cells differ in their adaptive response to injury and their potential to promote regeneration as well as their role in development of neuropathic pain [103]. The potential and involved mechanisms have been described extensively in reviews [21,26,104]. One difference between the myelinating and non-myelinating Schwann cells is that the latter express several biomarkers also found on developing Schwann cells (e.g., neural cell adhesion molecule (NCAM), p-75 neurotrophin receptor (p-75 NTR), glial fibrillar acidic protein (GFAP), and L1 NCAM) [86]. The relations between the myelinating and non-myelinating cells and the surrounding basal lamina are relevant for the regeneration process. Further players of the process, i.e., fibroblasts, blood vessels, and macrophages, are located and assembled in the endoneurium and participate in different ways during the regeneration process [105] (Figure 4). The myelinating nerve fibers are the key players during the regeneration process. They show a remarkable plasticity with a dependency on environmental signals during the degeneration and regeneration processes [105].
Figure 4Schematic drawing of an injured neuron, and the crucial cells involved in the degeneration and regeneration processes in the peripheral nerve with various components of the activities highlighted. After an injury, rapid and delayed injury signals are initiated, some advanced by axonal transport, eliciting upregulation of transcription factors in nerve cell body and Schwann cells with gene transcription. Wallerian degeneration is rapidly started, led by the invading macrophages, to clean up the myelin debris and fragmented parts of the axons. The extracellular matrix (ECM) is modified (i.e., “tissue niche”) through different mechanisms, thereby being beneficial for the Schwann cells that stimulate the regenerating axons. The direction of the outgrowing axons is guided by the formed sprouts with the growth cones at the tip (see details of growth cone in square with microtubules (green) in the sprout and actin filaments (red) in the filopodia), palpating the environment and approaching the target. Formation of blood vessels (angiogenesis), controlled by the extracellular environment [106], is crucial for the degeneration and regeneration processes. Several of the processes, including glycation of the tissue, are affected by diabetes. For details see text.
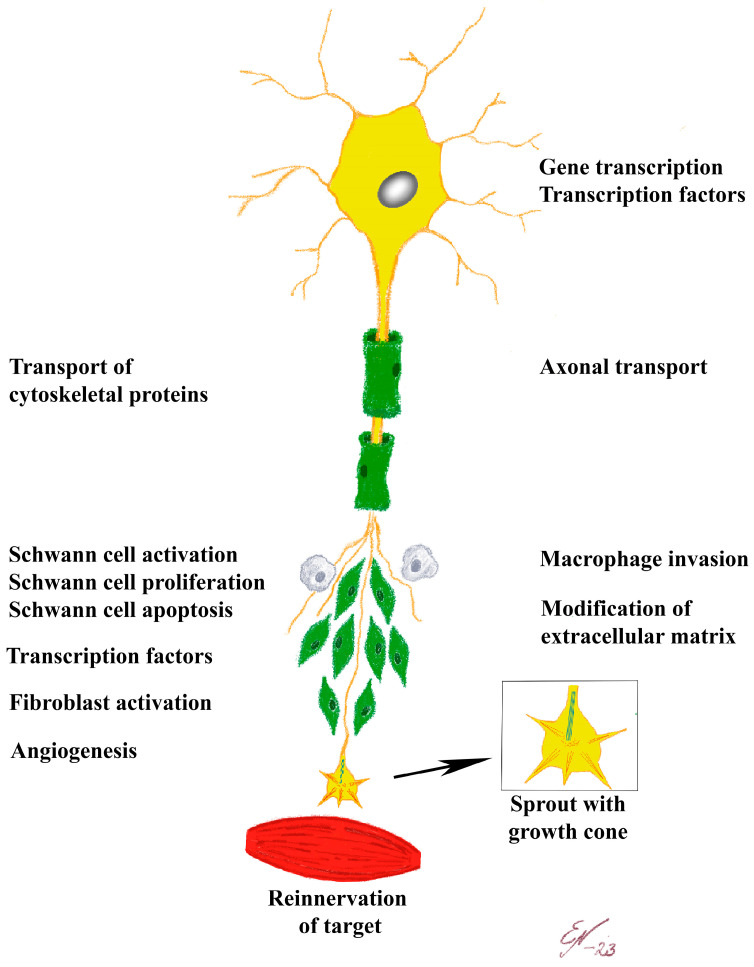


The extracellular matrix (ECM) proteins, such as laminin, fibronectin, and type IV collagen, impact the Schwann cells with a relevant role in remyelination after the nerve injury [107,108,109,110]. ECM proteins promote an early adhesion of Schwann cells in in vitro models in a different way, where laminin and fibronectin have a higher ability to induce the migration of Schwann cells [53,107,111,112]. In addition, laminin is also reported to be more relevant for inducing myelination and maturation of the myelin sheath with increased thickness of nerve fibers [107]. These in vitro experiments are supported by in vivo experiments showing that laminin has superior effect in myelin sheath formation and repair of the axonal structure, thereby resulting in a more optimal g-ratio (i.e., thickness of the myelin sheath) compared to that of fibronectin and collagen IV [107]. It has also been reported that nerve guidance conduits, which include multiple factors that improve nerve regeneration across large nerve defect after reconstruction, should be related to inflammation and angiogenesis, where the ECM components fibronectin, laminin 1, and laminin 2 in specific ratios are highlighted [54].

During the degeneration process, extensive alterations occur. Analysis of cell type-specific gene changes reveals that Schwann cells and endoneurial fibroblasts are crucial. The communication between these cells involves potential signals for the regeneration of blood vessels early in the process, along with the recruitment of macrophages, including T-cells [113], and the activation of the Schwann cells by macrophages [114] (Figure 4). The Schwann cell response at multiple timepoints early after a nerve crush has provided insight into changes in transcription in Schwann cells after a nerve injury [115]. The interaction between the Schwann cells, the nearby laying axons, and the intraneural blood vessels has been emphasized as a relevant factor in the development of diabetic neuropathy [83,116,117], and is of relevance for the ongoing regeneration and formation of the regenerative clusters [25].

The detailed mechanisms related to Schwann cell behavior after nerve injury and regeneration in diabetes are still not fully clarified, although differences have been observed concerning the expression of transcription factors and apoptotic factors [30,118,119] as well as how they interact with other key players [10]. However, Schwann cells in adipose tissue, including adipose-derived stem cells [120], may regulate the plasticity of the nerves in such a tissue [121]. There may be a dysregulation in diabetes, where there is a small nerve fiber demyelinating neuropathy observed in specific diabetic animal models. In these models, the alterations in the gene expression of Schwann cells are similar to those in human adipose tissue [121].

In clinical situations, it is crucial that axons can grow even if the nerve reconstruction or nerve transfer procedures have been performed with a delay; thus, the Schwann cells have a “best before date”. The ability of Schwann cells to be activated after denervation is impaired over time in healthy rats [122,123], but similarly, the denervation of the affected muscles may be even more relevant for functional outcome after delayed nerve reconstruction or nerve transfer surgery [124]. The ability of axonal outgrowth in diabetic Goto-Kakizaki rats is still preserved after a delayed nerve reconstruction as indicated in studies using chitosan conduits and autologous nerve grafts [125,126].

## 5. Macrophages and Schwann Cells in Nerves and in Dorsal Root Ganglia after Injury

The interplay between the responses of the macrophages and the Schwann cells is crucial for the degeneration and regeneration processes in health and disease (Figure 4). The importance of macrophage biology should be highlighted in the perspective of the resident and infiltrating macrophages in the nerve as well as their ability to affect the capability of sensory neurons to regrow. Macrophages, which invade the site of injury, the distal nerve end, and the dorsal root ganglia (DRG) after nerve injury, release a variety of factors. These factors include neurotrophic factors, such as leukemia inhibitory factor (LIF). The macrophages also contribute to the formation of synapses, beyond the well-described functions such as clearing myelin fragments and cellular debris at the site of injury and in the distal nerve end [39]. The chemokine CCL2 acts on the macrophage receptor CCR2 to facilitate the invasion of the macrophages after nerve injuries [43]. The resident macrophages consist of about 8% of the cells in the peripheral nerve and are important for the homeostatic state of the environment of the nerve. After the injury, additional monocyte-derived macrophages invade both the site of injury and the distal nerve end. There are two distinct populations of macrophages, which have been reported to have different functions. They are roughly classified as the proinflammatory M1 and the anti-inflammatory M2 macrophages, which may change during the regeneration process [39,43]. Despite extensive research on macrophages, their detailed functions are not fully described after nerve injury. Their behavioral phenotype as well as their function are described as a decidedly plastic process, which seems to vary as a function of time, location, and environment (see review by Wofford et al. [39]). Thus, there are probably many independent signals that attract and modify the behavior of the macrophages. This includes their temporal variations during the regeneration process [73], which may facilitate nerve regeneration [84].

The different phenotypes of macrophages seem to affect axonal clearance and axonal outgrowth at the site of injury and in the distal nerve end in different ways, while the mechanism(s) by which macrophages affect the nerve cell bodies are still not clarified completely [43]. The dedifferentiation process of the Schwann cells after injury to progenitor-like cells, which together with the resident macrophages and other inflammatory cells, contributes to the remodeling of the environment, thereby producing the conditions for an ideal axonal outgrowth after injury [73]. The macrophages may be a key factor for the guidance of Schwann cells along blood vessels, where the macrophage-derived vascular endothelial growth factor-A (VEGF-A) influences the polarity of the microvasculature in a nerve defect [127]. The Schwann cells move along the surface of polarized blood vessels across a nerve injury or a nerve defect, thereby guiding the Schwann cells and subsequently the accompanying axons. It has been proposed that hypoxia in the “bridge” between transected nerve ends potentiates the formation of the blood vessels, which is sensed by macrophages through the secretion of VEGF [73]. Again, this is a factor involved in the neuronal events occurring in diabetes [128]. This mechanism may be of relevance when larger nerve defects are bridged with autologous nerve grafts. These nerve grafts should be applied in a well-vascularized bed after a nerve reconstruction.

The interplay between macrophages, Schwann cells, the extracellular matrix with their proteins (i.e., tissue niche), microvessels, and axons are also essential in diabetes [116,129,130]. The relevance of macrophages for the impaired nerve regeneration described in some diabetic animal models, like streptozotocin(STZ)-induced diabetes with a high blood glucose level, should also be emphasized. The macrophages are described to be connected to a delayed clearance of the myelin debris in a complex manner, also including advanced glycation end products (AGEs) in the endoneurial ECM proteins [131]. ECM is also important in the pathophysiology of polyneuropathies [132]. The cell–matrix interaction is crucial for a variety of functions during degeneration and regeneration. These processes can be affected by the glycation of the extracellular matrix proteins caused by the increased blood glucose levels. The receptor for AGEs (i.e., RAGE) is a multiligand receptor of the immunoglobulin superfamily prominent in activated Schwann cells and invading macrophages after a nerve injury. It interacts with AGEs as well as with different proinflammatory and regulatory molecules (see review by Sango et al. [131]). As expected, the expression of RAGE is increased in diabetes. It has been suggested that an impaired axonal outgrowth in diabetic mice may be ascribed to a RAGE-dependent alteration of macrophage polarization. In addition, proinflammatory M1 macrophages are increased in diabetic wild-type mice compared to non-diabetic ones. A decrease and an increase in M1 and M2 macrophages, respectively, have been found in RAGE-deficient diabetic mice (see review by Sango et al. [131]). Thus, the biology of macrophages and their actions in nerve injuries and in neuropathies are crucial. The behavior of different macrophages may be a further research direction with the intention to improve axonal outgrowth after trauma as well as in different conditions like diabetes [39,43,129].

## 6. Genes and Proteomics in Human and Animal Nerves after Injury and in Diabetes

Most studies of the early and late responses of Schwann cells and other central players in the nerve degeneration and regeneration processes have focused on the events in animals. Few studies have studied the protein response in injured human peripheral nerves [133,134,135] or in humans with type 1 and type 2 diabetes [85,136]. It has been reported that two major genes, associated with repair Schwann cells, c-Jun and p75^NTR^, are up-regulated in injured human nerves in healthy subjects, while the myelin-associated transcription factor (EGR2) is down-regulated [135]. In a proteomics study of the sural nerve that was subjected to a nerve injury (surgery in patients with idiopathic Parkinson’s disease in connection with nerve grafting into substantia nigra; control nerve and then distal nerve end harvested at two weeks follow up; age 50–70 years), 8/20 most prominently up-regulated pathways were associated with gene expression through translation, while the top down-regulated pathways were associated with cytoskeletal organization or structure as well as to neuron and synapses [133]. Detailed changes were reported concerning growth factors (e.g., glial cell-derived neurotrophic factor; GDNF, brain-derived neurotrophic factor; BDNF; transforming growth factor-β; TGF-β and VEGF), myelination, Schwann cell differentiation (e.g., c-Jun), regulation of apoptotic processes, and response to “axonal injury” [133,137]. The expression of the previously mentioned genes declines during a long-term denervation in accordance with an observed impaired regeneration over time after a nerve injury in experimental animals [49,50,51,122,135,138,139]. The corresponding response in subjects with diabetes is not clarified. Schwann cells in degenerated nerves, including those from humans, have two important functions, i.e., myelin debris clearance and antigen presentation via MHCII. These functions indicate that debris clearance occurs via phagocytosis-related mechanisms and type II immune regulation in addition to the relevant up-regulation of Jun [134]. The transcription factor c-Jun is essential for a Schwann cell response after injury and is also of relevance in aging [42,102]. A fast and substantial, although transient, increase in activated JNK (i.e., p-JNK) in the nuclei of healthy sensory neurons after injury is followed by the phosphorylation of c-Jun and the induction of ATF3 [140]. Thus, such a JNK-mediated activation of c-Jun is one of the first responses in the cell body after a nerve injury. This activation, together with the subsequent induction of ATF3, is connected to the outgrowth of the axons [140]. The response of ATF3 after nerve injury and repair may differ between healthy and diabetic rats depending on the diabetic rat model (e.g., Goto-Kakizaki rats and diabetic BB rats), which may explain differences in axonal outgrowth [30,118]. The response of these factors is complex since they may be involved in the pathophysiology of different neuropathies. c-Jun is aberrantly expressed in a few acquired and inherited neuropathies in humans [1]. There may also be an aberrant activation of a programmed axonal death in polyneuropathies, including diabetic neuropathies [1]. The timing of the activation of the molecular pathways in the Schwann cells after a nerve injury as a response to a specific nerve injury and how it regulates axonal degeneration and regeneration mechanism(s), including epigenetic and epitranscriptomic regulation [4], however, have not been fully delineated, particularly in diabetes and other conditions that may cause neuropathies (see details in Arthur-Farraj and Coleman 2021 [1]).

Analysis of the human proteome at the subcellular level is of crucial importance for understanding how neurons and Schwann cells respond to a nerve injury in accordance with what has been reported from other models [141,142]. An interesting question is also if there is a difference in proteomics of injured sensory and motor nerves after injury, including diabetes, where several proteins have been identified to be differentially expressed in injured motor and sensory nerves [143]. The outcome of nerve repair in human motor and sensory nerves differs, and its predictors include age, gender, repair time, repair materials, nerve injured, length of the defect, and duration of follow-up [144]. There is a lack of clinical studies that evaluate the outcome of nerve repair or reconstruction in diabetes. Analysis of proteomics in diabetes, in which de- and regenerative events are ongoing [23,24,25], has been recently presented; however, it is complex to analyze due to the substantial number of identified proteins. Many heat shock proteins, like HSP27, relevant in diabetic neuropathy have been detected in human nerve biopsies [85,136]. However, HSP27 has not been shown to be associated with axonal outgrowth in healthy rats or in diabetic Goto-Kakizaki rats after nerve injury and repair, despite an increase after injury, a finding which may be more related to survival [119].

## 7. Nerve Regeneration in Males and Females and in Diabetes

Although most nerve injuries in humans affect males, few studies have concentrated on investigating nerve regeneration mechanisms in male and female rats or mice in clinically relevant injury and repair and reconstruction models [30,145,146,147,148]. Gender-specific perspectives are relevant in view of reported enhancement of axonal outgrowth in males, and sustained synaptic inputs by exercise are observed in male and female rats, but the requirements of exercise protocols differ between sexes [30,145,146,147,148]. In contrast to studies examining gender effects on nerve regeneration after a nerve crush, axonal outgrowth seems to be better in healthy and diabetic *male* rats than in corresponding healthy and diabetic *female* rats after a sciatic nerve injury and direct repair [30]. Some gender differences, as well as differences depending on diabetic status, in activated Schwann cells are reported, while number of apoptotic Schwann cells does only differ depending on diabetic status [30]. Interestingly, the patterns of activated and apoptotic Schwann cells at the site of the lesion and in the distal nerve end are to some extent more complex, indicating the importance of the environment for the outgrowing axons [30]. In a diabetic Goto-Kakizaki model, the blood glucose level correlates positively to axonal outgrowth as well as to the number of activated Schwann cells at the site of the lesion and in the distal nerve end, where the number of activated Schwann cells are balanced against the number of apoptotic Schwann cells (i.e., cleaved caspase 3-stained cells) [30]. These studies indicate that there are gender differences in nerve regeneration in healthy and Goto-Kakizaki rats (i.e., resembling type 2 diabetes) after a sciatic nerve injury and direct repair [30]. Using the same diabetic rat model, axonal outgrowth is enhanced compared to the outgrowth in healthy rats when a 10 mm sciatic nerve defect is reconstructed with either a hollow chitosan conduit or an autologous nerve graft, particularly in the latter [149]. In such nerve regeneration models, the expression pattern of activated (ATF3-stained) and apoptotic (cleaved caspase 3-stained) Schwann cells is complex at the site of the injury and in the distal nerve end [149]. Again, the preoperative glucose level positively correlates with axonal outgrowth after the used reconstruction methods bridge such a nerve defect. Overall, the data from these Goto-Kakizaki-rat models with direct nerve repair of the nerve injury or reconstruction of a nerve defect also indicate that the vascularity of the environment in which the outgrowing axons advance is relevant. Thus, a moderately increased blood glucose level can compensate and nourish the axons, Schwann cells, and other key players, when the axons regenerate in an avascular environment, i.e., a nerve graft or a hollow conduit.

The diabetic BB-rat model, expressing much higher blood glucose levels than the dia-betic Goto-Kakizaki-rat model, displays a different regeneration pattern after a nerve injury and repair. This is because a higher number of ATF3-stained Schwann cells as well as cells expressing cleaved caspase 3 [118]. ATF3 is a relevant marker to detect an increased susceptibility to nerve compression in diabetes as seen in diabetic BB-rats [150]. Direct repair after nerve injury provides a better axonal outgrowth in healthy rats than in the clinically relevant diabetic Goto-Kakizaki-rat model. However, the moderately increased blood glucose level in diabetic Goto-Kakizaki-rats may positively influence the regeneration capacity in experimental nerve reconstruction models (i.e., autologous nerve grafts and hollow chitosan conduits) used to bridge nerve defects. Thus, in these nerve regeneration models, the axons grow in an initial avascular environment, in contrast to a direct nerve repair, where the distal nerve end is vascularized if the nerve repair is performed without tension [151]. These studies indicate that during certain circumstances there is a difference in nerve regeneration and in susceptibility to nerve compression between healthy and diabetic rats depending on the diabetic rat model. Thus, it is important to use different nerve injury and regeneration models in animals, including diabetic rat models [152], to understand nerve regeneration mechanisms. Therefore, one may extrapolate these mechanisms during decision making in a clinical situation.

The transcription factors, with subsequent effects, may be altered, as indicated above, in a hyperglycemic environment [30,61,149,153,154], or may be less highlighted in the literature for hypoglycemic conditions, leading to a high risk of degeneration with limitations to regeneration [155]. The response of nerve regeneration and expression of transcription factors as well as of apoptotic markers to hyperglycemia may depend on the used experimental diabetic rat model as well as the applied nerve injury models [30,118,149]. In a neuropathy with high blood glucose level, Schwann cells may be severely affected by the induction of apoptosis [156], which may be related to the activation of Schwann cells [118]. Heat shock proteins are other substances discussed in relation to nerve injury [119], nerve compression disorders [157], and neuropathy [74,158,159]. The heat shock protein HSP27, which seems to be relevant for the prevention of development of neuropathy in humans [158] and is the main highlighted protein among heat shock proteins, is up-regulated after a nerve injury in both the injured nerves and the dorsal root ganglia [119]. The expression is affected by the interaction between axons and Schwann cells [119]. Interestingly, in animal models, an increased expression of HSP27 is observed at the uninjured side of sciatic nerves in diabetic Goto-Kakizaki rats [119]. In such a model, the decreased axonal outgrowth seems to be associated with diabetes, i.e., in a model with modest increase in blood sugar levels, which is in accordance with other models with much higher blood sugar levels [29,118]. However, the increased expression of HSP27, neither in the sciatic nerve nor in DRG, has any impact on axonal outgrowth [119].

The discussion about the timing of nerve reconstruction is also relevant in a diabetic condition. It is reported that an impaired nerve regeneration is inflicted by a delayed reconstruction after a transection or laceration injury in healthy rats [51,102,122,160,161,162,163]. The process seems to not be impaired further in a diabetic condition, as has been shown after reconstruction of a nerve defect using immunomodulatory chitosan nerve guides [125]. As indicated above, the nerve regeneration process is better in autologous nerve grafts than in hollow chitosan conduits. Surprisingly, it is enhanced in diabetic Goto-Kakizaki rats compared to healthy rats after a nerve reconstruction, which may be related to the moderately increased blood glucose [149]. This was despite a higher percentage of ATF3 immunostained cells in nerve segments from healthy rats than in diabetic Goto-Kakizaki rats, but it may be related to the complexity in view of the presence of apoptotic (i.e., cleaved caspase 3-stained) Schwann cells in the distal nerve end in the diabetic rats [149]. As indicated, the expression of phosphorylated extracellular-signal-regulated kinase 1/2 (i.e., p-ERK 1/2) is increased in a peripheral nerve after nerve injury, particularly in Schwann cells [61]. Diabetic rats (i.e., STZ-induced diabetic and BB-rats, which are similar) show a lower intensity of expression of p-ERK 1/2 both proximal and distal to a nerve injury compared to healthy rats, which may be a factor of relevance for the different regeneration capacity in diabetes [61]. The specific problems of nerve regeneration in a diabetic condition should also be associated with early glycation and advanced glycation end products (AGE) in the extracellular matrix proteins. This glycation of laminin and fibro-nectin was reported to be the reason for the failure of sensory nerve regeneration in STZ-induced diabetes [164], one of the models that usually induces high blood glucose levels [152].

## 8. Conclusions and Significance

The most important objective in nerve degeneration and regeneration mechanisms after nerve injury and in neuropathy is to understand how genes and different factors are activated through well-orchestrated programs [165,166,167]. However, we still must understand how the processes are orchestrated at the cellular level. This understanding will stem from detailed structural analysis of nerve biopsies with nanotomography. Therefore, appropriate animal models, both for health and disease, are relevant. The understanding of the normal biology and the physiology of the neurons and the other key player cells under normal conditions and under stress are of crucial importance [168,169]. Animal studies should be combined with studies that also involve a variety of analyses in humans, e.g., diffusion tensor imaging (DTI), diffusion kurtosis imaging (DKI), and tractography [170] as well as neurophysiology [171]. A future direction of research in axonal degeneration and regeneration, focusing on regeneration in diabetic conditions, should continue to reveal the various steps and stages in the different processes. Findings from this research, in turn, should lead to the development of pharmacological therapies as an adjunct to surgical repair of injured nerves. There are several early critical events occurring both in the axons and their related Schwann cells, as well as in macrophages, during the regeneration process from the site of injury during the initial phase, where a delayed nerve repair and reconstruction may influence the outcome (see reviews [12,51,52,102,163]). Critical evaluation of data from different diabetic rat models is important for the regeneration process after using various nerve repair and reconstruction prototypes. A moderately increased blood glucose level may be beneficial for the outgrowing axons, the migration of the Schwann cells, the modification of the extracellular matrix, and other key players in an initial avascular environment before revascularization, thus forming an optimal “tissue niche”. Even if we have an advanced knowledge about the relationship between the outgrowing axons and the Schwann cells as well as the delicate orchestrated mechanisms during the process, we still need to understand the different aspects of the process. It is important to understand the dynamic processes that are constantly occurring in neuropathies, like diabetic neuropathy, where both degeneration and regeneration phenomena occur [25]. The advances in the technology and bioinformatics have led to an increased understanding of the behavior of different cell types, and genomics, transcriptomics, and proteomics can be used to obtain integrated information about the events occurring in injured nerves and in nerves subjected to neuropathy [74,172,173]. Furthermore, using single-cell-transcriptional profile analysis of different cells, for instance, it may be possible to observe the heterogeneity among the major players in the peripheral nerves, including Schwann cells, the neuron, the fibroblast, the macrophages, endothelial cells, and other key players, where there might be both inter- and intra-organ differences [174].

## Figures and Tables

**Figure 1 ijms-24-15241-f001:**
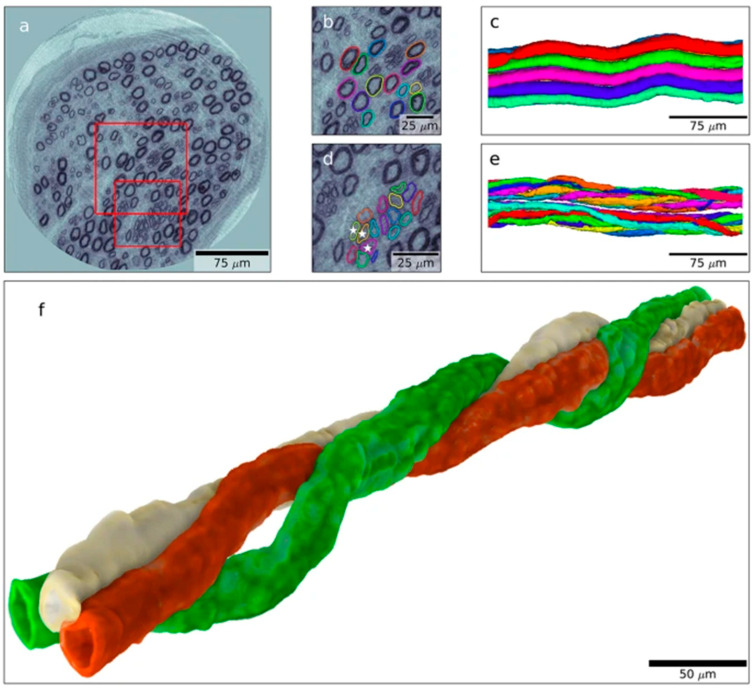
Nanotomographic images of a human posterior interosseous nerve from an individual with diabetes. (**a**–**c**) Tomographic slices with enlarged areas used to create 3D images in (**c**) of the segmented nerve fibers marked in (**b**) (healthy nerve fibers with different colors; note wave form). In (**d**), an enlargement of an area marked by small red square in (**a**) is shown with regenerative clusters (i.e., regenerating nerve fibers indicated by different colors) shown in 3D in (**e**). Finally, 3D volume rendering the three axons is marked by stars in (**d**) and are illustrated in (**f**) with one axon (green) twisting around the other two (grey and red). Reproduced by permission from Dahlin et al. [25] (based on and published under a CC BY License: http://creativecommons.org/licenses/by/4.0/, accessed on 5 October 2023).

**Figure 2 ijms-24-15241-f002:**
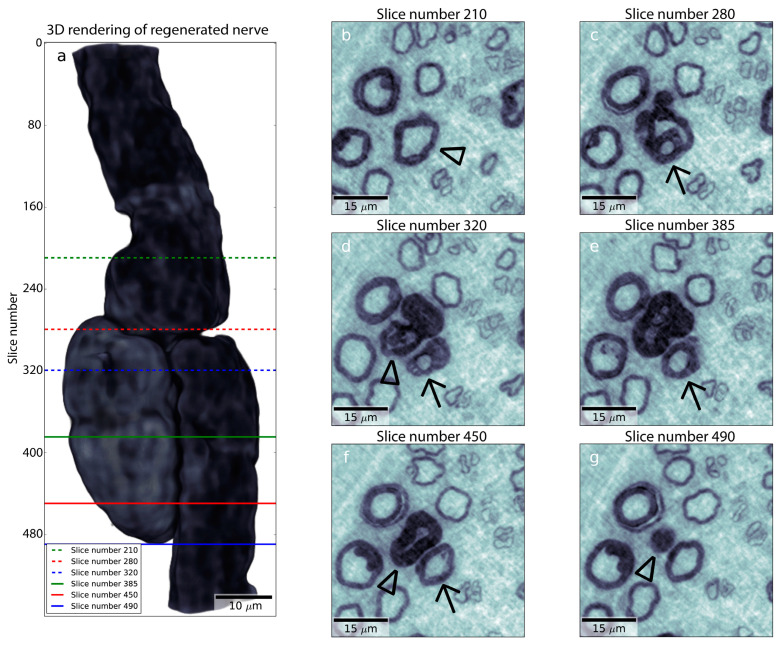
Visualization of a regenerative event in a similar nerve biopsy as in Figure 1. 3D rendering in (**a**) showing only an abnormal axon, where the regenerative event is shown in panels (**b**–**g**) and is indicated by lines in panel (**a**). Arrow indicates regenerating axon. Arrowhead indicates the original axon. Reproduced by permission from Dahlin et al. [25] (based on and published under a CC BY License: http://creativecommons.org/licenses/by/4.0/, accessed on 5 October 2023).

**Figure 3 ijms-24-15241-f003:**
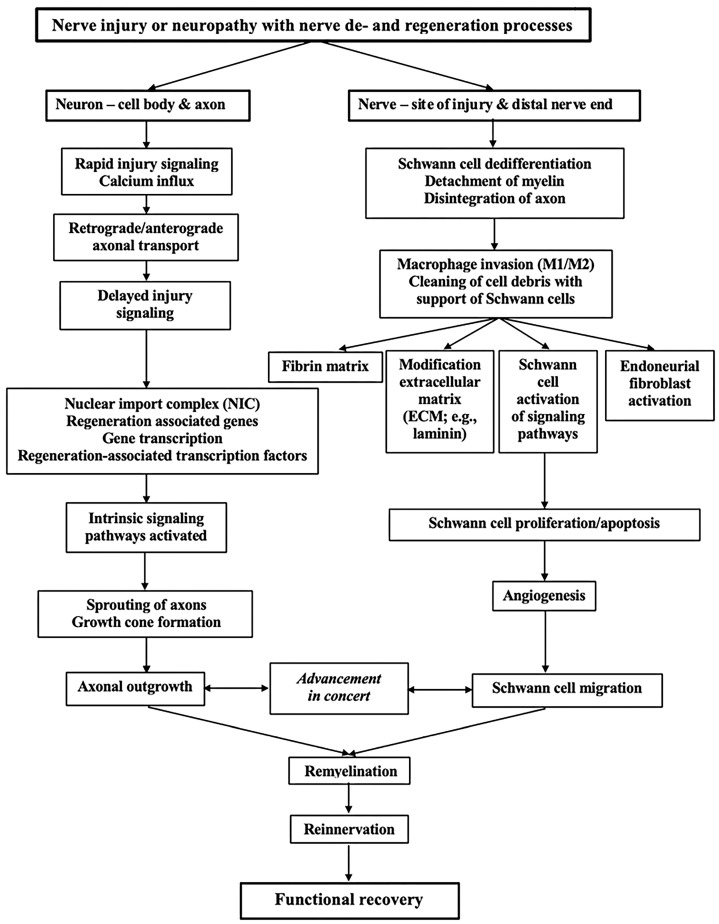
Schematic ray diagram of some crucial events that occur in an injured nerve or in neuro-pathy in which degeneration and regeneration processes appear with a focus on the neuron (left) and the other relevant alterations that occur at the site of injury and at the distal nerve end (right). The intention of the events is to convert the cellular machinery from a maintenance to a production type, i.e., nerve regeneration with the final goal of reinnervation and functional recovery (see text for details).

## Data Availability

Not Applicable.

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
