# Peer review of "The Dynamics of Nerve Degeneration and Regeneration in a Healthy Milieu and in Diabetes"

_ijms, 2023, doi:10.3390/ijms242015241_

Round 1
Reviewer 1 Report
This narrative review summarizes what is known about the processes of peripheral nerve degeneration and regeneration in response to injury. The writing is clear, and the organization is helpful. The discussion of growth factors and other molecules/pathways in this process is also useful to the reader. Overall, this work is very well done. I had two small questions/clarifications on what was written:
- Is there any data about why diabetes seems to improve nerve regeneration in the animal models? Does this correlate at all to clinical experience?
- I also recommend summarizing the role of diabetes in nerve degeneration/regeneration dynamics in the conclusion section (very briefly, likely ~1 sentence), since this is a major focus of the manuscript based on the title.
Overall, the English usage is excellent. There are a few minor errors in grammar/word choice, but these do not impair the readability of the manuscript.
Author Response
Question: This narrative review summarizes what is known about the processes of peripheral nerve degeneration and regeneration in response to injury. The writing is clear, and the organization is helpful. The discussion of growth factors and other molecules/pathways in this process is also useful to the reader.
Reply: Thank you!
Overall, this work is very well done. I had two small questions/clarifications on what was written:
Question: Is there any data about why diabetes seems to improve nerve regeneration in the animal models? Does this correlate at all to clinical experience?
Reply: Thank you for a good question. It is not easy to answer since nerve regeneration has been evaluated in different diabetic animal models. As pointed out in one of the models a moderately increased blood sugar level is beneficial in nerve grafting, but not of any benefit after a direct nerve repair; thus, depends on the vascularity of the distal nerve end. Writing further clarified in the text (section 7).
Question: I also recommend summarizing the role of diabetes in nerve degeneration/regeneration dynamics in the conclusion section (very briefly, likely ~1 sentence), since this is a major focus of the manuscript based on the title.
Reply: Thank you for a good point; sentences are added in conclusion.
Question: Comments on the Quality of English Language:
Overall, the English usage is excellent. There are a few minor errors in grammar/word choice, but these do not impair the readability of the manuscript.
Reply: Thank you; I have read the manuscript carefully and revised the language, including shortening of sentences.
Reviewer 2 Report
In this review the author discussed the degeneration and regeneration processes are focused on events going on in healthy as well as in a “diseased” microenvironment, using diabetic neuropathy as the experimental model. They discussed genes and different factors are activated through the well-orchestrated programs in neurons and it is important to understand during nerve regeneration intended for axonal outgrowth, migration of Schwann cells along suitable substrates, invasion of macrophages, appropriate conditioning of extracellular matrix, fibroblast activation, etc. and activation of other players in healthy and diabetic conditions. Additionally, they also highlighted the importance the dynamic processes, constantly occurring in neuropathies, like diabetic neuropathy, with concomitant degeneration and regeneration, which requires advanced technology and bioinformatics.
Overall, the manuscript was well written. However, a few concerns/comments needed to be explained/modified.
- Line 32 Can you explain this term for the common reader of your MS “Wallerian degeneration”
- Line 42 7 References are cited at once it would be better to cite few and explain it well.
- Lone 43 please classify whether the authors want to discuss neurological disorders linked to axon or diabetes.
- Line 70 in hyperglycemia in this condition regeneration happened and please clarify if the degeneration does not happen.
- ‘Line 107-111 please avoid long sentences.
- Line 169-171 it would be nice if the authors provided a ray diagram that proposed the changes in injury and regeneration
- Line 212-214 It would be nice if the authors explained well here
- Line 231-233 What does it mean? Please explain it in a better way
- Line 245-259 Figures are needed in the manuscript for clear understanding of the pathway involved in this
- Line 360-361 How these the authors could describe it for increasing both M1 and M2
- Line 366 I think the section should be only focused on diabetes, it would be better
- Line 378-383 The authors did not discuss diabetes in this section
- Line 391-392 This section should be well explained.
- Line 400 What were the authors views regarding this
- Line 416-419 The authors could prepare a table to show the differences if possible.
- Line 451 Only this HSP27 did not involve any other heat shock proteins.
- Line 501 is important to cite references in the conclusion part, already there were many.
Author Response
Overall, the manuscript was well written. However, a few concerns/comments needed to be explained/modified.
Questions:
- Line 32 Can you explain this term for the common reader of your MS “Wallerian degeneration” Reply: The term Wallerian degeneration has been clarified.
- Line 42 7 References are cited at once it would be better to cite few and explain it well. Reply: Changes have been made in the text, where the individual references has been highlighted with their message.
- Lone 43 please classify whether the authors want to discuss neurological disorders linked to axon or diabetes. Reply: Sentence rewritten for clarifications.
- Line 70 in hyperglycemia in this condition regeneration happened and please clarify if the degeneration does not happen. Reply: Sentences clarified.
- Line 107-111 please avoid long sentences. Reply: Sentences changed.
- Line 169-171 it would be nice if the authors provided a ray diagram that proposed the changes in injury and regeneration. Reply: A ray diagram is inserted in this section (Figure 3).
- Line 212-214 It would be nice if the authors explained well here. Reply: A number of sentences have been added in this paragraph with an additional reference.
- Line 231-233 What does it mean? Please explain it in a better way. Reply: Sentences clarified.
- Line 245-259 Figures are needed in the manuscript for clear understanding of the pathway involved in this. Reply: It is not possible to create any figure concerning these substances and how they influence the remyelination. Sentences have been modified.
- Line 360-361 How these the authors could describe it for increasing both M1 and M2. Reply: Sentences have been changed/clarified.
- Line 366 I think the section should be only focused on diabetes, it would be better. Reply: Sentences added in this section about diabetes.
- Line 378-383 The authors did not discuss diabetes in this section. Reply: Sentences and references added in this section with more details about diabetes.
- Line 391-392 This section should be well explained. Reply: Sentences and references added in this section with more details about diabetes.
- Line 400 What were the authors views regarding this. Reply: Changes with clarifications and correction of reference are made.
- Line 416-419 The authors could prepare a table to show the differences if possible. Reply: The expression of ATF3 and cleaved caspase 3 at the site of injury and in the distal nerve end after the two used nerve reconstruction models are difficult to visualize based on the referenced article. I have tried to clarify the sentence without adding details. The reader is referred to the original article.
- Line 451 Only this HSP27 did not involve any other heat shock proteins. Reply: Mainly HSP27 has been the one that is highlighted as a protective substance although around 32 different HSPs have been found in peripheral nerves. Clarifications added.
- Line 501 is important to cite references in the conclusion part, already there were many. Reply: The intention was to add a number of key references in this context. References adjusted.
Reviewer 3 Report
the review is generally well-written and organized, however, it misses illustrative figures to summarize the pathways involved in axonal regeneration as well as the genes related to injury happening to nerves
minor errors
Author Response
Question: The review is generally well-written and organized, however, it misses illustrative figures to summarize the pathways involved in axonal regeneration as well as the genes related to injury happening to nerves
Reply: I have added a “ray diagram” and a schematic figure (Figure 3 and figure 4) with the key players regarding involved cells. In the former I have included briefly different steps, but a figure with the substantial number of involved pathways for signaling after the injury and for the regeneration processes is difficult to create; maybe better to refer to the large number of referenced articles.
Question: Comments on the Quality of English Language: minor errors
Reply: See above; the manuscript has been carefully read and errors corrected.